# Does imputation matter? Benchmark for predictive models

**Katarzyna Woźnica** [1]   **Przemyslaw Biecek** [2] [1]

## Abstract

Incomplete data are common in practical applications. Most predictive machine learning models do not handle missing values so they require some preprocessing. Although many algorithms are used for data imputation, we do not understand the impact of the different methods on the predictive models' performance. This paper is first that systematically evaluates the empirical effectiveness of data imputation algorithms for predictive models. The main contributions are (1) the recommendation of a general method for empirical benchmarking based on real-life classification tasks and the (2) comparative analysis of different imputation methods for a collection of data sets and a collection of ML algorithms.

## 1. Introduction and related works

In practical tasks in data analysis and machine learning, one of the most common problems are missing values in collected data. On the other hand, many established machine learning algorithms require fully observed data sets without any missing entries. Due to, imputation is a necessary step in preprocessing and the subject of handling with missing data is a challenge for practitioners. We can observe this for example on the Kaggle platform where users compete in real-life machine learning tasks. Competitors often share their knowledge among other approaches to imputation data. This gives us insight into the trend of applied methods: there is a tendency to apply simple methods such as mean or mode replacement regardless of their limitations.

At the same time, many statisticians work on more complex methods with theoretical foundations. Rubin (1976) was the first to formalise the universal three categories of the process generating: missing completely at random (MCAR), miss-

ing at random (MAR) and missing not at random (MNAR). Since then numerous techniques of substituting missing data were developed. In R package wide range of single imputation are implemented: missForest (Stekhoven & Bühlmann), softImpute (Hastie et al., 2014), VIM (Kowarik & Templ, 2016), missMDA (Josse & Husson, 2016) . A variety of multiple imputation techniques are also available: mice (van Buuren & Groothuis-Oudshoorn, 2011), Amelia (Honaker et al., 2011), missMDA (Josse & Husson, 2016). Most of these implementations deal to impute missing entries in continuous and categorical variables. In addition, most of these packages enable more than one method of imputation. In the platform R-miss-tastic (Mayer et al., 2019) can be found a comprehensive summary of existing techniques.

Due to the plenitude of available packages and methods, global evaluation of existing techniques is desirable. So far only a few articles address this necessity (Kyureghian et al., 2011; Jadhav et al., 2019). These comparisons focused on the quality of imputed data. They considered simulated data and applied several imputation methods was assessed in terms of the accuracy of predicting the missing values.

In most cases handling missing data is prepossessing step to complete data before primary modelling task. For practitioners, a crucial aspect in choosing of imputation method is the impact of the selected procedure on predictive power on the ML model. Recently, in machine learning more attention is paid to benchmarking and comparison of various predictive algorithms or importance of hyperparameters but only a few papers took into account selection of prepossessing techniques. Brown & Kros (2003) provide a descriptive study of imputation impact on machine learning algorithms but did not support these conclusions with any empirical results. Hutter et al. considered substitute missing values with mean, median or mode as one of the hyperparameters in importance analysis, but imputations were limited to simple ad-hoc approaches and their impact was inappreciable.

In this paper, we research into the contributions of imputation methods to the improvement of predicting power of machine learning classification algorithms. We provide a benchmark on 13 real-life tasks and face simple methods with more sophisticated ones. Proposed benchmark has universal nature and can be applied to assessment influence of any imputation methods on a wide range of data sets and

---

[1]Faculty of Mathematics and Information Science, Warsaw University of Technology [2]Faculty of Mathematics, Informatics and Mechanics, University of Warsaw. Correspondence to: Katarzyna Woźnica <k.woznica@mini.pw.edu.pl>.

*Presented at the first Workshop on the Art of Learning with Missing Values (Artemiss) hosted by the $37^{th}$ International Conference on Machine Learning (ICML).* Copyright 2020 by the author(s).

*Table 1.* Statistics of considered OpenML data sets: number of instances, percentage of missing values, number of continuous variables, number of continuous variables with missing values, number of categorical variables, number of categorical variables with missing values.

| dataset name (dataset ID) | # obs | prc of missings | # numeric | # numeric w. missings | # categorical | # categorical w. missings |
|---|---|---|---|---|---|---|
| ipums_la_99-small (1018) | 8844 | 7% | 15 | 0 | 41 | 14 |
| adult (1590) | 48842 | 1% | 4 | 0 | 9 | 3 |
| eucalyptus (188) | 736 | 3.9% | 14 | 10 | 2 | 0 |
| dresses-sales (23381) | 500 | 14.7% | 1 | 1 | 12 | 9 |
| colic (27) | 368 | 16.3% | 5 | 5 | 15 | 13 |
| credit-approval (29) | 690 | 0.6% | 6 | 2 | 10 | 5 |
| sick (38) | 3772 | 2.2% | 6 | 6 | 22 | 1 |
| labor (4) | 57 | 33.6% | 8 | 8 | 9 | 8 |
| SpeedDating (40536) | 8378 | 1.8% | 59 | 58 | 64 | 3 |
| hepatitis (55) | 155 | 5.4% | 6 | 5 | 14 | 10 |
| vote (56) | 435 | 5.3% | 0 | 0 | 17 | 16 |
| cylinder-bands (6332) | 540 | 5.1% | 19 | 18 | 15 | 7 |
| echoMonths (944) | 130 | 7.5% | 6 | 6 | 4 | 1 |

algorithms.

## 2. Experiments settings

Reliable source of real-world data sets is OpenML, especially collection of classification task OpenML100 (Bischl et al., 2017). In this experiment, we focus on binary classification and pick data sets with at least one column with missing values. We select from the OpenML database thirteen data sets meeting these criteria. In Table 1 we present summary of basic information about considered tasks. Every data set was split into two part: training data frame consisting of 80% of instances and test data set. ML models were learnt on the training part and then metrics were validated on test data.

### 2.1. Imputation methods

We select seven methods of handling missing data. For some data, some methods did not work. Next to imputation name in bracket we give a number of data sets which particular methods succeed in imputing on. We test two simple ad-hoc methods: **random** (13) - every missing entry was replaced with a value drawn independently from observed values of considered feature, **mean** (13) - filling missing values with mode for categorical variables and mean for continuous variables of complete values in a feature. Moreover, we consider four single imputation methods. The first is **softImpute** (10) - for numeric variables fit a low-rank matrix approximation to a matrix with missing values. For categorical variables missing values are imputed with mode. The second is **missForest** (11) - imputation with predictions of random forest model, trained on complete observations. Available for both numeric and categorical variables, From VIM package we choose two methods: **VIM kknn** (13) - k-Nearest Neighbour imputation can be applied to numeric and categorical features, and **VIM hotdeck** (13) - sequential, random hotdeck algorithm. As representative of multiple imputation,

we include **mice** (10) - we test default methods from this package: predictive mean matching for numerical variables and polytomous logistic regression for categorical ones.

Some of the above-mentioned methods depend on additional parameters. In this benchmark we used default values. Reproducible scripts are available at `https://github.com/ModelOriented/EMMA`. We also tried Amelia and missMDA but they failed to impute most data sets, and they were excluded from the benchmark. To prevent data leakage and simulate a real-world application, we should fit imputation methods on train data set and then apply this on the test sample. Unfortunately, in used packages implementations this is impossible so we decided to impute data separately on train and test data. For the same reasons, the target variable was excluded from the imputation step.

### 2.2. ML Algorithms

We select five types of algorithms which should capture the different structure of data: logistic regression with regularization (implemented in glmnet package), classification tree (rpart), random forest (implemented of ranger package), k-nearest neighbours and xgboost. In this benchmark we leave aside hyperparameter tuning, every algorithm was trained with default settings.

In the first step for every data sets on train and test part, missing values are substituted with seven imputation methods. Then on train data five types of algorithms were fit and on test data we reported value of two performance measures obtained on test data: Area Under Curve (AUC) and F1. We consider two types of measures because of that some data sets have imbalanced response variable and F1 captures this aspect of quality of performance.

# 3. Results

Because selected measures are incomparable across data sets, next to the comparison of values of metrics we create the ranking. For every task and every machine learning algorithms we rank imputation methods, scores can range from 1 to 7. The higher and better measure the lower rank obtain this imputation technique. Methods which did not work on specific task get the lowest score (7). For some data sets and algorithms, despite different imputation methods, some models achieve exactly the same measure values. In case of ties we assign a maximum value of ranks, so rank 1 corresponds to evidently the best model.

In our analysis, we focus on three main questions about globally the best imputation method and the interaction between imputation and classifiers. We check trends in obtained results and attempt to draw conclusions about the optimal workflow for incomplete data.

## 3.1. Does exist the best universal imputation method?

In Figure 1 for every imputation methods we show the distribution of ranks based on F1 and AUC measure. Single score corresponds to one task and one algorithm, so for every method there are 65 scores. We can interpret this figure in various ways. If we assume that the best methods are to achieve rank 1 to 3 most frequently, then *mean* substituting is the winner for F1 measure and *kknn* method for AUC measure. On the other hand, every imputation methods give the best measure of F1 and AUC for at least one task and algorithm.

For both measures top positions are taken by simple methods as *random*, *mean* or *kknn* from VIM package. Against this background arising question whether these methods work effectively on similar tasks and ML algorithms or rather oppositely. In Figure 2 we present percentage of covered best results in rankings of F1 and AUC measure by single imputation methods and all pairs of them. As single imputation we would choose *random* and *mean* or *kknn* for F1 and AUC respectively. At the same time, combinations of two methods work definitely better and they are able to cover above 50% of best results. For F1 measure *missForest* and *random* methods cover outstanding percentage of best methods. Three of the most effective methods for F1 are *random*, *missForest* and *VIM_hotdeck*, but *missForest* imputation result in improvement of coverage of AUC best results. For AUC measure optimal pair is *mean* and *VIM_kknn* substituting but *mean* and *missForest* are very close to them and, what is more, achieve better results for F1 measure. We may notice that *missForest* imputation takes high positions for both measures.

To extend this approach we perform a greedy search to find the optimal sequence of imputation methods covering a wide

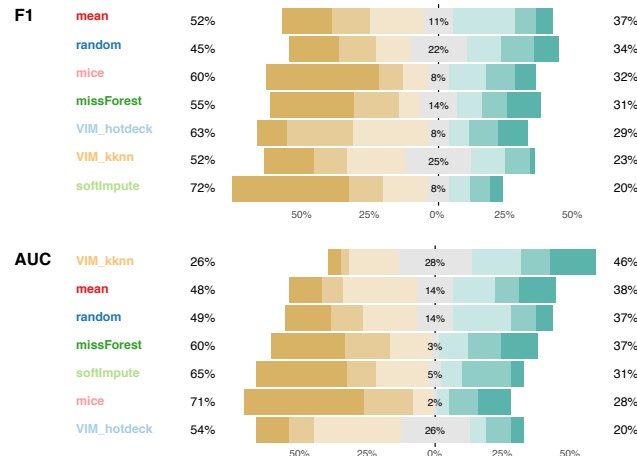

Figure 1. Bars describe how often a given method of imputation had the best results (rank 1, dark-green) or the worst results (rank 7, dark-orange) for a particular pair ML-model/dataset. The top ranking is based on F1 measure while the bottom one is based on AUC. The percentages on the right describe how often a method was in position 1 to 3. The percentages on the left describe how often a method was in position 5 to 7.

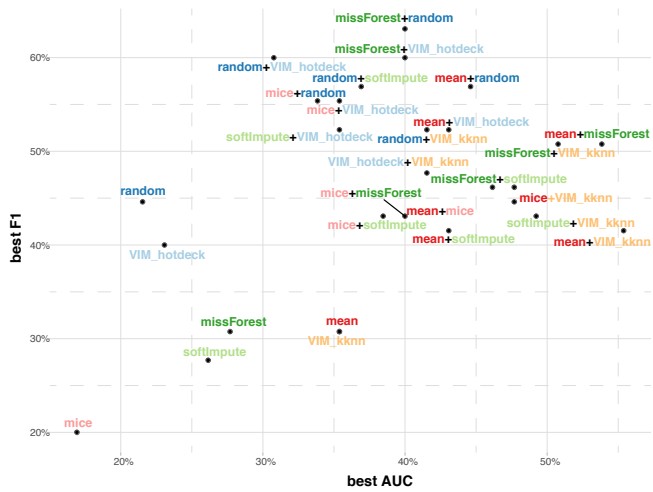

Figure 2. The OX axis shows how often the indicated imputation method has the best results measured by the AUC. The OY axis shows how often the indicated imputation method has the best results measured by F1. The points marked A+B refer to the better of the two indicated methods (parallel max).

range of tasks and algorithms. In the Figure 3 on the left panel we on the OY axis we see these sequences for F1 and AUC respectively. We see that for F1 *random*, *missForest* and *hotdeck* covers above 75% of optimal imputation methods for combinations of data set and ML model. For AUC the first two positions are the same as in the Figure 2, but the third is *mice* imputation. It suggests that this imputation method works complementarily to simple approaches.

As we see, shown results may be concluded in different ways depending on the considered measure and there is no

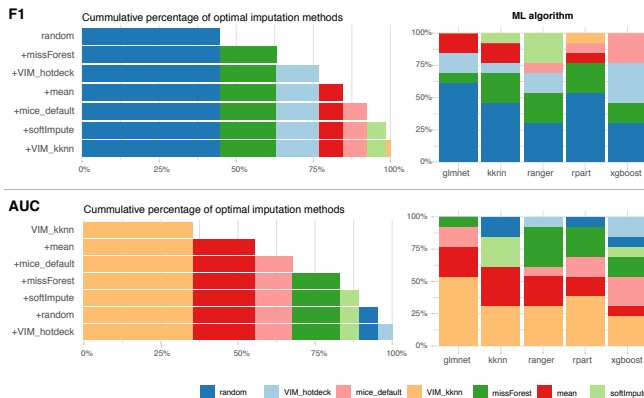

*Figure 3.* Left panel: On the OY axis there are consecutive imputations from the greedy search for F1 and AUC respectively. On OX axis there is a cumulative percentage of tasks and ML models for which one of imputation method was optimal. Right panel: Contribution of subsequent imputations from greedy search broken down by ML algorithms. On the OY axis there is a cumulative percentage of covered the best imputation.

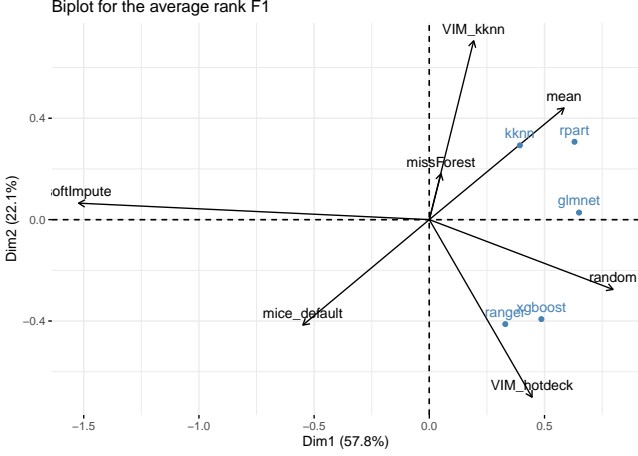

*Figure 4.* Biplot for a PCA made for an average of the rankings for pairs imputation-method / ML-method. The first coordinate correlates with the average ranking, the best results have the method mean. The second coordinate reveals the method's preferences. The mice method works better for the ranger model than for the kknn model. See Table 2 for details.

one answer to the question about universal the best imputation method. For considered tasks, it is difficult to select imputation technique maximizing both measures. Next question arising from this analysis is an interaction between imputation methods and type of classifier algorithm.

### 3.2. Does exist the best imputation method for machine learning algorithm?

In Table 2 we present averaged across data sets ranking based on F1 measures for imputation methods and classifier

*Table 2.* Average rank for a particular method of imputation and method of construction of the classifier

|  | glmnet | kknn | ranger | rpart | xgboost |
|---|---|---|---|---|---|
| mean | 4.38 | 3.85 | 4.77 | 4.69 | 4.62 |
| mice_default | 5.54 | 4.92 | 4.85 | 5.15 | 4.23 |
| missForest | 5.15 | 4.38 | 4.85 | 4.62 | 4.46 |
| random | 4.46 | 4.00 | 4.00 | 4.77 | 4.31 |
| softImpute | 5.69 | 4.69 | 4.92 | 5.92 | 5.46 |
| VIM_hotdeck | 4.85 | 4.62 | 4.15 | 4.85 | 3.92 |
| VIM_kknn | 5.15 | 4.23 | 5.08 | 4.15 | 4.69 |

model. The lower score indicates better methods. According to averaged ranking, *mean* imputation is the best in 2 out of 5 models, for glmnet and kknn model. For ranger ML-method, *random* imputation wins but *missForest* takes top position in rpart model. For xgboost, *hotdeck* achieves on average best score. Deeper insight into interaction of imputation and classifiers gives principal component analysis (PCA) performed on averaged rankings in Figure 4. The first PCA coordinate positively correlates with averaged ranking so *mean* method gives the best results. Second coordinate reveals model preferences. *Mean*, *missForest* and *VIM_kknn* methods cooperate with rpart and kknn while *mice* works with ranger and xgboost. This conclusion goes along with Figure 3 on upper right panel where we present results for greedy search of optimal set of imputations splitting by ML models. We see that for ranger, xgboost and rpart models*mice* provide enhancement of substituting missing values in relation to *random* and *missForest* imputation.

## 4. Conclusions

To our best knowledge, this is the first empirically benchmark of imputation methods in terms of their impact on the predictive power of classifier algorithms. This kind of verification of proposed methods enables a better understanding of the pros and cons of imputation techniques. We proposed a general plan of the experiment which can be extended to different data sets, imputation methods and predictive algorithms. In our experiment, simple imputation methods achieve surprisingly good results but we can not conclude that more advanced methods should be given up. We focus on the impact on their predictive power but methods with statistical foundations achieve better results in accuracy in imputed values.

Included analysis of results do not provide single universal default the best imputation method even for a particular ML-model. What is more, the selection of these imputation methods is sensitive to the considered performance measure. We observe some trends in results but generally their structure is very complex. For human is very difficult to summarise this in a concise way. This is the area to employ meta-learning model to capture these high-level interactions. In future work, we aim to extend this analysis with adding

new data sets and training surrogate model to deeper understanding the complex interactions between the structure of missing data in tasks, the assumption of imputation method and ML algorithms (Woźnica & Biecek, 2020). Another extension may be considering hyperparameters optimization in machine learning models as well as imputation methods.

## Acknowledgements

This work was financially supported by the NCN Opus grant 2017/27/B/ST6/01307.

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
