# OpenReview forum: "Does imputation matter? Benchmark for real-life classification problems."
_ICML.cc/2020/Workshop/Artemiss — ICML Artemiss 2020_

### Official Review · AnonReviewer2 · 2020-06-18
**Does imputation matter? Benchmark for real-life classification problems.**

**Rating:** 5
**Confidence:** 4

**Review:**

Several methods are discarded even though they are methods that have shown good performance in many situations.
Moreover, we do not know why some methods are better than other on certain datasets.  It is very surprising that average and random are the 2 most interesting methods.
The authors should give insights on which methods to use according to the circumstances or the kind of dataset.

---

### Decision · Program_Chairs · 2020-07-02

**Decision:**

Accept

**Comment:**

We're happy to accept this paper at Artemiss. We'll contact you soon to inform you about more details concerning the format of your presentation at the workshop, and the camera-ready version deadline. Please take into account the referee's comments to write the camera-ready version.